# Effects of Different Carbon Sources on Fumonisin Production and *FUM* Gene Expression by *Fusarium proliferatum*

**DOI:** 10.3390/toxins11050289

**Published:** 2019-05-22

**Authors:** Yu Wu, Taotao Li, Liang Gong, Yong Wang, Yueming Jiang

**Affiliations:** 1Key Laboratory of Plant Resource Conservation and Sustainable Utilization, Guangdong Provincial Key Laboratory of Applied Botany, Key Laboratory of Post-harvest Handling of Fruits, Ministry of Agriculture, South China Botanical Garden, Chinese Academy of Sciences, Guangzhou 510650, China; wuyu@scbg.ac.cn (Y.W.); taotaoli@scbg.ac.cn (T.L.); lianggong@scbg.ac.cn (L.G.); 2School of Life Sciences, University of Chinese Academy of Sciences, Beijing 100039, China; 3Zhongshan Entry-Exit Inspection and Quarantine Bureau, Zhongshan 528403, China; gbhg5674@customs.gov.cn

**Keywords:** *Fusarium proliferatum*, fumonisin biosynthesis, carbon source, environmental stress, gene expression

## Abstract

*Fusarium proliferatum* can infect many crops and then produce fumonisins that are very harmful to humans and animals. Previous study indicates that carbon sources play important roles in regulating the fumonisin biosynthesis. Unfortunately, there is limited information on the effects of carbon starvation in comparison with the carbon sources present in the host of fumonisin production in *F. proliferatum*. Our results indicated that *F. proliferatum* cultivated in the Czapek’s broth (CB) medium in the absence of sucrose could greatly induce production of fumonisin, while an additional supplementation of sucrose to the culture medium significantly reduced the fumonisin production. Furthermore, cellulose and hemicellulose, and polysaccharide extracted from banana peel, which replaced sucrose as the carbon source, can reduce the production of fumonisin by *F. proliferatum*. Further work showed that these genes related to the synthesis of fumonisin, such as *FUM1* and *FUM8*, were significantly up-regulated in the culture medium in the absence of sucrose. Consistent with fumonisin production, the expressions of *FUM* gene cluster and *ZFR1* gene decreased after the addition of sucrose. Moreover, these genes were also significantly down-regulated in the presence of cellulose, hemicellulose or polysaccharide extracted from peel. Altogether, our results suggested that fumonisin production was regulated in *F. proliferatum* in response to different carbon source conditions, and this regulation might be mainly via the transcriptional level. Future work on these expressions of the fumonisin biosynthesis-related genes is needed to further clarify the response under different carbon conditions during the infection of *F. proliferatum* on banana fruit hosts. The findings in this study will provide a new clue regarding the biological effect of the fumonisin production in response to environmental stress.

## 1. Introduction

Mycotoxins as secondary metabolites produced by some fungi are capable of causing disease and even death in humans and other animals [1]. Fumonisins (FBs) were first discovered in the late 1980s and received worldwide attention due to their toxicity for humans or animals [2]. In humans, FB has been shown to be well correlated to a high incidence of esophageal cancer in South Africa [3]. The FBs consists of FB_1_, FB_2_, FB_3_ and FB_4_, of which FB_1_ is the major component [4]. On the basis of available toxicological evidence, the World Health Organization’s International Program known as the International Cancer Research Institute released a reference of carcinogen list in 2017 and classified fumonisin B_1_ (FB_1_), fumonisin B_2_ (FB_2_) and fusarium C as class 2B carcinogens. FB_1_ have the largest proportion of the total fumonisins accounting for up to 70% while FB_2_ and FB_3_ usually make up 10%–20% of the total fumonisin content [5]. FB_1_ mainly contaminated maize and its products [5], leading to two diseases [6] which occur in domestic animals: equine leukoencephalomalacia and porcine pulmonary edema syndrome. The two diseases involve disordered sphingolipid metabolism and cardiovascular disease [7,8]. The products from *Fusarium verticillioides* and *Fusarium proliferatum* are their main sources. Fumonisins B (FBs) are major mycotoxins synthesized by *F. proliferatum* [9]. *F. proliferatum* colonizes a broad range of host plants, such as maize, wheat, asparagus, banana, and various conventional crops or profitable crops [10,11]. In these hosts, root rot disease [12,13] and black point symptoms [14] have been caused by infection of *F. proliferatum*. Moreover, *F. proliferatum* can produce various toxic secondary metabolites, such as fumonisins [15], beauvericin [16], fusaric acid [17], and nontoxic secondary metabolites like bikaverin [18]. 

The biosynthetic pathway of FBs has been well documented. Fumonisins consist of a 19 to 20 carbon aminopolyhydroxyalkyl chain that generally undergoes four steps: reductions of carbonyl compounds, hydroxylation, alanine condensation, and esterification of two tricarboxylic acids [19]. Moreover, the gene cluster (*FUM*) of the fumonisin biosynthesis has been identified in *F. proliferatum* [20]. However, the *FUM* does not contain a pathway-specific regulatory gene, unlike other fungal secondary metabolite gene clusters [21]. In addition, several genes appear to regulate fumonisin biosynthesis such as a *Zn(II)2Cys6* DNA binding protein (*ZFR1*), which is not linked to the *FUM* but rather encodes a polypeptide with significant homology to fungal proteins that contain a DNA binding motif consisting of a *Zn(II)2Cys6* binuclear cluster [22,23]. Additionally, the possible fumonisin biosynthesis mechanism was also reported [24]. Carbon sources including sucrose and glucose have been proven to affect greatly fumonisin production [25]. Meanwhile, sugar is one of the nutrition components in edible fruits and can provide the main carbon source for fungi during their infection process. Stepien et al. [26] reported that *F. proliferatum* strains from different hosts are genetically diverse while host plant extracts can change the expression patterns of *FUM* and fumonisin production.

Banana, as one of the most economically important fruit crops worldwide, deteriorates easily due to rot development caused by postharvest pathogens, including *F. proliferatum* [25]. Banana fruit as a hosts for these types of pathogens can provide nutrition for *F. proliferatum* infection, especially the banana peel is rich in sugar ingredients like cellulose, hemicellulose and pectin as carbon sources when the fruit becomes edible. *F. proliferatum* might face these polysaccharides while infecting banana fruit. Additionally, banana peel is the first natural infection place for *F. proliferatum*, and, thus, *F. proliferatum* mainly grows on the banana peel after infection. Hence, investigation of the effect of the polysaccharides from banana peel on the fumonisin production from *F. proliferatum* might enable better understanding of the interaction between banana fruit and *F. proliferatum*.

Recent research suggests that *Colletotrichum spp.*, *Alternaria alternate* and *Fusarium oxysporum* could alkalize the host plant for better infection. In contrast, a *Penicillium* spp. can acidify the host environment for attack [27,28,29]. In addition, carbon availability in the environment is a key factor for triggering the host pH change [30]. As is known, one of the most significant changes is a rapid increase of sugar content during fruit ripening. Unfortunately, there is currently a lack of research on the response of mycotoxin biosynthesis in a pathogen with the carbon status of the host plant and carbon stress. In this study, we investigated the production of fumonisin in the absence of sucrose and an addition of sucrose under the lack carbon condition in *F. proliferatum*. Additionally, the fumonisin production in *F. proliferatum* by cellulose, hemicellulose or polysaccharide extracted from banana peel instead of sucrose was also investigated. The expressions of these genes involved in fumonisin biosynthesis were conducted to understand the underlying mechanism. This study might provide new information on the regulation of fumonisin biosynthesis in response to different nutrition environments of *F. proliferatum*.

## 2. Results

### 2.1. Effect of Different Sucrose Conditions on the Growth, Sporulation and FB_1_ Content of F. Proliferatum 

The *F. proliferatum* strain was cultivated in different culture media while mycelial growth rate, sporulation and FB_1_ production were checked after 3 and 6 days. Colony morphology varied when *F. proliferatum* was cultured in different media (Figure 1). In short, the *F. proliferatum* showed a better mycelial growth rate and more sporulation in the medium in the presence of sucrose than the medium in the absence of sucrose. Furthermore, an additional sucrose supplementation recovered the growth of *F. proliferatum*, which was almost the same as the medium in the presence of sucrose (Figure 2A,B). For FB_1_ content, the study indicated that *F. proliferatum* can produce FB_1_ in the medium with or without sucrose, but FB_1_ content in the medium without sucrose was significantly (*p* < 0.05) higher than the culture medium with sucrose, while an additional supplementation of sucrose after 3 days of culture in the medium without sucrose significantly (*p* < 0.05) inhibited the FB_1_ production (Figure 2C) according to the Duncan’s multiple comparison in ANOVA analysis. Thus, supplementation of sucrose recovered the growth but also inhibited the FB_1_ production by *F. proliferatum*, suggesting that the sucrose starvation increased FB_1_ production. 

### 2.2. Effect of Different Carbon Sources on the Growth, Sporulation and FB Content of F. Proliferatum

After the *F. proliferatum* strain was cultivated in the culture media containing five different carbon sources, mycelial growth rate, sporulation and FB content were investigated after 3 and 6 days. A different colony morphology of *F. proliferatum* was observed when five carbon sources were used (Figure 3) while the growth rate was almost the same as that on the 6th day (Figure 4A), but the sporulation of *F. proliferatum* was induced in the media containing cellulose, hemicellulose, and polysaccharide extracted from banana peel (Figure 4B). Furthermore, the contents of FB_1_ and FB_2_ produced by *F. proliferatum* in the culture media with cellulose, hemicellulose and polysaccharide extracted from banana peel were significantly (*p* < 0.05) lower than that in the culture medium with sucrose according to the Duncan’s multiple comparison in ANOVA analysis (Figure 4C,D).

### 2.3. Effect of Different Carbon Sources on the Expressions of FB-Related Genes

Figure 5, Figure 6 and Figure 7 present the results of the expressions of *FUM* and *ZFR1* of the *F. proliferatum* cultured with different carbon media. The expressions of these genes were clearly induced in the medium in the absence of sucrose, with 1–5 folds higher than those in the medium with sucrose. After supplementation of sucrose to the medium in the lack sucrose, the expression levels of these genes were reduced significantly (Figure 5). Furthermore, these genes were significantly (*p* < 0.05) reduced in the culture media containing cellulose, hemicellulose (Figure 6), or polysaccharide from banana peel according to the Duncan’s multiple comparison in ANOVA analysis (Figure 7). These results further confirmed that the lack of sucrose greatly induced the expressions of *FUM* and *ZFR1*. 

## 3. Discussion

Contamination by fumonisins is an important issue that affects crop quality and human health. To control FB production, attention has been paid to these key factors that greatly affect the synthesis and its mechanisms. Previous studies have exhibited that environmental and abiotic factors, such as carbon source, nitrogen source and pH, greatly influenced the FB biosynthesis of *F. proliferatum* [31,32]. In the present study, we comparatively evaluated the effects of lack carbon source and additional carbon sources such as sucrose, cellulose, hemicellulose and polysaccharide from banana peel on the growth and FB biosynthesis of *F. proliferatum*. The Appendix A present polysaccharide information obtained from banana peel. Our results exhibited that the microconidia growth generally increased in the presence of cellulose, hemicellulose or polysaccharide extracted from banana peel (Figure 4B) but decreased in the absence of sucrose (Figure 2B). The previous study also demonstrated that a sufficient carbon source is beneficial for mycelial and conidia growth [33]. In contrast, FB_1_ production of *F. proliferatum* was induced in response to lack carbon stress but it was inhibited in the culture medium containing carbon source, which was in agreement with the result of Kohut et al. [34], who reported that nitrogen starvation stress induced *FUM* expression and increased fumonisin production in *F. proliferatum*. It is interesting to note that FB_1_ or FB_2_ production from *F. proliferatum* cultivated in the culture medium containing the polysaccharide extracted from ripe banana peel was lower than unripe banana peel. The result may be due to the difference in the degradation of polysaccharide during banana fruit ripening. Thus, carbon sources played a key role in growth and fumonisin biosynthesis of *F. proliferatum*. Importantly, carbon starvation encouraged the FB production of *F. proliferatum*, but its mechanism needs to be elucidated further.

Fungal pathogens are able to modulate environment pH to increase their infective potential [35]. In the case of carbon excess pathogens can induce acidification while, in contrast, alkalization occurs under carbon deprivation condition [28]. During banana fruit ripening, starch is degraded gradually into sucrose [36], and sugar can be oxidized into carbon dioxide through tricarboxylic acid, whereas an enhanced glycolysis rate and production of organic acids help to secrete metabolites that decrease the host pH value, which could result in activation of some genes to enable fungi to use a specific set of pathogenicity factors to infect the host [30]. Moreover, the host pH environment could affect the production of FBs in *F. proliferatum*. These findings exhibit a high biological relevance because *F. proliferatum* infection may undergo a transition from alkalization to acidification as the sugar contents gradually increase during the ripening of banana fruit. In addition, the production of FBs produced by *F. proliferatum* was lower in the culture medium containing polysaccharides extracted from ripe peel than from an unripe peel (Figure 4C,D). These results were consistent with the report of Li et al. [37], who found that the production of FBs produced by *F. proliferatum* was significantly inhibited under the acidification condition.

Previous research indicated that mycotoxin biosynthesis could be mainly regulated at the transcriptional level [38]. To further investigate the possible mechanism of different carbon sources involving in FB biosynthesis, we examined the expression profiles of these related genes related to the FB biosynthesis pathways. The expressions of the crucial FB biosynthesis-related genes were affected greatly by various environmental factors [39]. Considering the FB pathway in *F. proliferatum*, real-time reverse transcription PCR (RT-PCR) assays were used. *FUM1*, *FUM3*, *FUM6, FUM8*, *FUM15*, *FUM18* and *FUM19* belonging to the member of *FUM* cluster includes 17 genes, as designated to be *FUM1*, *FUM2*, *FUM3*, *FUM6*, *FUM7*, *FUM8*, *FUM10*, *FUM11*, *FUM13*, *FUM15*, *FUM18* and *FUM21*, respectively [40,41,42]. In this study, the expressions of these genes in *F. proliferatum* demonstrated a positive relationship with the FB production under different carbon conditions. For example, when *F. proliferatum* was cultured in the medium without sucrose, FB_1_ content was significantly induced with significantly up-regulated expressions of these genes. In particular, *FUM1* encodes a polyketide synthase in the early step of participating the assembly of the FB backbone, while *FUM8* is an aminotransferase gene which catalyzes the formation for a biologically active FB_1_ molecule. A previous study confirmed the positive relationship between the expression of these two genes and FB production [25]. In addition, *ZFR1* encodes DNA-binding proteins containing a zinc binuclear cluster (*Zn(II)2Cys6*) belonging to the Gal4p family of transcriptional factor, regulates diverse pathways and acts as a positive regulator of FB_1_ biosynthesis in *F. proliferatum* [43,44]. *FUM15* and *FUM18* encoding cytochrome P450 monooxygenases and longevity assurance factor, respectively, were reported to be correlated with the FB production [41]. In the present study, the induced expressions of all these genes were in agreement with increased FB production when *F. proliferatum* was cultured in the medium in the absence of sucrose, while, in contrast, when additional supplementation of sucrose to the medium occurred, these gene expressions were significantly decreased in association with the reduced FB_1_ content. Jayashree and Subramanyam (2000) reported that some stress factors greatly affected mycotoxin production from fungi [45]. For example, FB_1_ production from *F. proliferatum* was enhanced by nitrogen starvation stress [34]. Therefore, carbon starvation stress can mediate the regulation of FB biosynthesis in *F. proliferatum*. Moreover, when *F. proliferatum* was cultured in the medium containing polysaccharide from ripe banana peel, the FB_1_ and FB_2_ contents and the expression levels of *FUM* cluster were significantly lower than the medium with unripe banana peel, which was in agreement with the previous report which indicated that a sufficient carbon source was only beneficial for fungal growth [33]. 

In general, our results showed that a carbon resource greatly influenced fungal growth and secondary metabolites. Based on the present results, we hypothesized that the changed fumonisin production might be a response of *F. proliferatum* to nutrition environmental stress to help to infect banana fruit host, which needs to be investigated further.

## 4. Conclusions

In this study, the different carbon sources significantly affected the FB biosynthesis in *F. proliferatum.* Results exhibited that *F. proliferatum* can regulate the FB production in response to different nutrition conditions while the regulation was performed via the transcriptional level. Importantly, *F. proliferatum* enhanced the FB biosynthesis with increased expression levels of *FUM* cluster and *ZFR1* in the absence of sucrose. In addition, when *F. proliferatum* was cultured in the medium containing the polysaccharide extracted from unripe banana peel, the higher contents of FB_1_ and FB_2_ in association with the increased expression levels of *FUM* gene cluster were obtained and compared with the medium containing the polysaccharide extracted from ripe banana peel. Future work on the expressions of the FB biosynthesis-related genes is needed to further clarify the infection ability of *F. proliferatum* on banana fruit as a host.

## 5. Materials and Methods

### 5.1. Fungal Strain and Growth Condition

The strain of *F. proliferatum* was originally isolated from decayed banana fruit and was routinely maintained in the laboratory on potato dextrose agar (PDA) (Oxoid, Basingstoke, Hampshire, England) at 28 °C. The spores were washed from the PDA plate with sterile water, and then conidia were counted with a hemocytometer and then diluted to a concentration of 1 × 10^7^ conidia/mL before 2 mL of the diluted spore suspension was inoculated to the Czapek’s broth (CB) medium. The medium was prepared for culture of *F. proliferatum* according to the method of Li et al. [46]. Conical flasks (250 mL) were prepared, containing 100 mL of the CB medium (3.0 g/L NaNO_3_, 1.0 g/L K_2_HPO_4_, 0.5 g/L MgSO_4_·7H_2_O, 0.5 g/L KCl and 0.01 g/L FeSO_4_) supplemented with 30 g/L sucrose (Aladdin, Shanghai, China), cellulose, hemicellulose (FeiBo, Guangzhou, China), or polysaccharide extracted from ripe or unripe banana peel, as shown in following Section 5.2, and then sterilized for 20 min at 121 °C. The conical flasks were incubated at 28 °C with 200 rpm and were shaken in the dark for sporulation, fumonisin and molecular analyses. 5 μL of conidia suspension was inoculated on CB plates with supplemented with 1.5% agar and then used for the morphological and growth assessments. Three biological replicates were conducted.

### 5.2. Polysaccharide Extraction

Ripe fruit of banana (*Musa acuminate* L. AAA group, cv. Brazilian) with a fully yellow skin and unripe fruit at harvested were obtained from a commercial orchard in Guangzhou, China. Banana peel tissues were collected, frozen with liquid nitrogen and smashed into powder with a pulverizer, respectively. Polysaccharides were extracted by the method of John et al. [47] with some modification. Briefly, 100 g of power from twelve banana fingers was homogenized with 1 L of distilled water and then incubated at 105 °C for 2 h. The extract was filtered through gauze and then concentrated by a vacuum rotary evaporator (Eyela N1100 V-W, Tokyo Rikakikai Co. Ltd., Tokyo, Japan). Anhydrate ethanol was added into the extract to obtain a final concentration of 60% (*v*/*v*) and then maintained for 12 h at 4 °C. The obtained precipitate was dissolved in distilled water and the solution was dialyzed against running tap water for 24 h. Finally, the solution was lyophilized to obtain the crude polysaccharides [48].

### 5.3. Fumonisin Analysis

Fumonisin was extracted from 20 mL of the liquid culture filtrate according to the method of Jian et al. [49]. The ABSCIEX triple quad^TM^ 5500 UPLC–MS/MS system (AB SCIEX, Framingham, MA, USA) accompanied by a Ekspert 100 UPLC column (C18 column, 100 × 2.1 mm, 3 µm particle size, Thermo, USA) was used to analyze the FB_1_ and FB_2_. The FB_s_ analyses were conducted according to the method described by Li et al. [37]. Briefly, 10 µL of sample was injected for fumonisin analysis. An optimized gradient of mobile phase (A: acetonitrile and B: 5 mM ammonium acetate) were applied as follows: the initial composition of the mobile phase 10% of A/90% of B was kept constant for 0.5 min, then the A solvent was linearly increased to 50% in 7.5 min. Finally, the A solvent was linearly decreased to 10% in 0.5 min and kept constant for 1 min. The flow rate of the mobile was 0.4 mL/min. Positive ionization was selected for mass spectrometric (MS) detection. A multiple reaction monitoring (MRM) function was employed for quantification, with the fragment ions at *m*/*z* 722.5 for FB_1_ and 706.4 for FB_2_, respectively. Three biological replicates were conducted.

### 5.4. RNA Isolation and cDNA Synthesis

For RNA isolation, the mycelia of *F. proliferatum* grown in the CB media with different carbon sources were filtered by Buchner funnel and then liquid nitrogen was added immediately and ground into powder. The powder (100 mg) was weighed and then used for RNA extraction. The RNA was extracted using the Hipure Fungal RNA Mini Kit (Magen, Guangzhou, China). The cDNA was synthesized using the cDNA PrimeScript^TM^ RT Master Mix Takara Kit (TAKARA-RR036A, Dalian, China). 

### 5.5. Expression Analysis by Real-Time Quantitative PCR

ABI7500 fast real-time fluorescence quantitative PCR instrument (Applied Biosystems, Foster City, CA, USA) was used for RT-qPCR assay using our previous method [50]. SYBR Premix Ex TaqTM mix (TaKaRa, Dalian, China) was used in this study with 20 µL of reaction system, including 10.0 µL of SYBR Premix Ex TaqTM, 0.4µL of PCR forward primer (10 µM), 0.4 µL of PCR reverse primer (10 µM), 0.4 µL of ROX reference dyeⅡ and 2 µL (20 ng) of cDNA. After amplification (40 cycles at 95 °C for 30 s, 95 °C for 5 s and 60 °C for 34 s), the relative expression levels of target genes were calculated using the formula 2^−ΔΔCT^ with *Histone H3* as the reference gene. All these genes were selected due to their important roles in fumonisin production and their responses to different carbon sources [25]. The following prime pairs designed by PrimerPremier 5 (PREMIER Biosoft International, Palo Alto, CA, USA), were used for RT-qPCR, as shown in Table 1. Three biological replications were conducted.

### 5.6. Statistical Analysis

All experiments were performed in triplicate. Data for each sample were statistically analyzed using SPSS software (Version 16.0, SPSS Inc., Chicago, IL, USA). One-way analysis of variance (ANOVA) followed by the Duncan’s multiple comparison was used for statistical significance analysis. Differences were considered to be significant at *p* < 0.05.

## Figures and Tables

**Figure 1 toxins-11-00289-f001:**
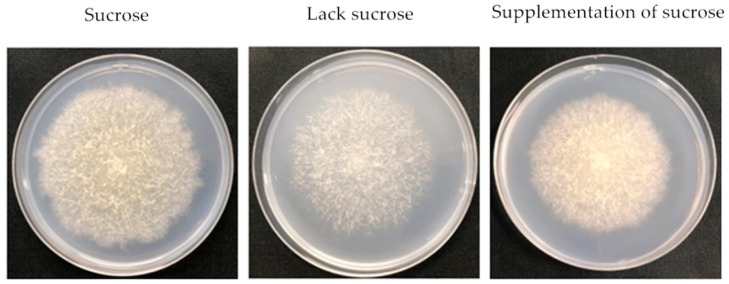
The morphology of *F. proliferatum* on CB media with different carbon sources. A.B.C.

**Figure 2 toxins-11-00289-f002:**
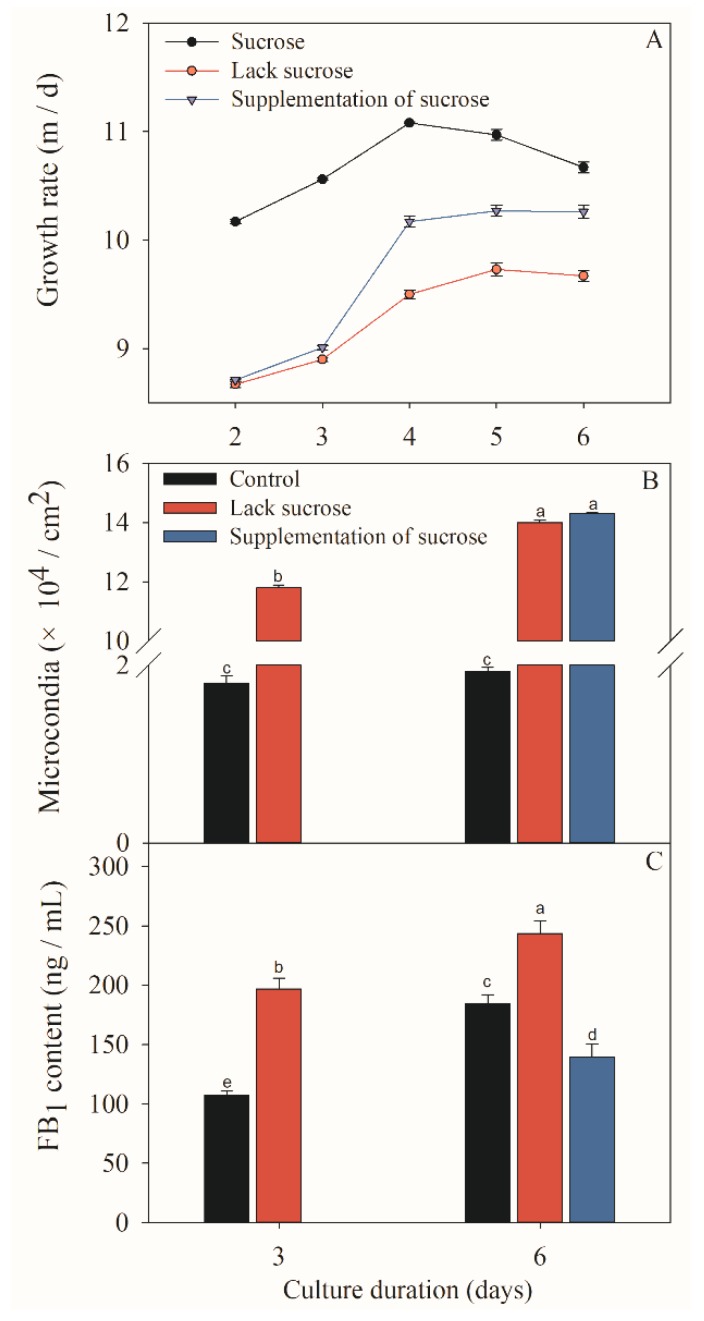
Effects of different sucrose conditions on fugal growth (**A**, **B**) and FB_1_ production (**C**) of *F. prolifeatum*. Sucrose: 6 days in the presence of sucrose; lack sucrose: 6 days in the absence of sucrose; and supplementation of sucrose: 3 days in the absence of sucrose, followed by 3-day culture after the supplementation of sucrose. The vertical bars indicate standard errors of three replicates. Different letters represent significant differences (*p* < 0.05).

**Figure 3 toxins-11-00289-f003:**
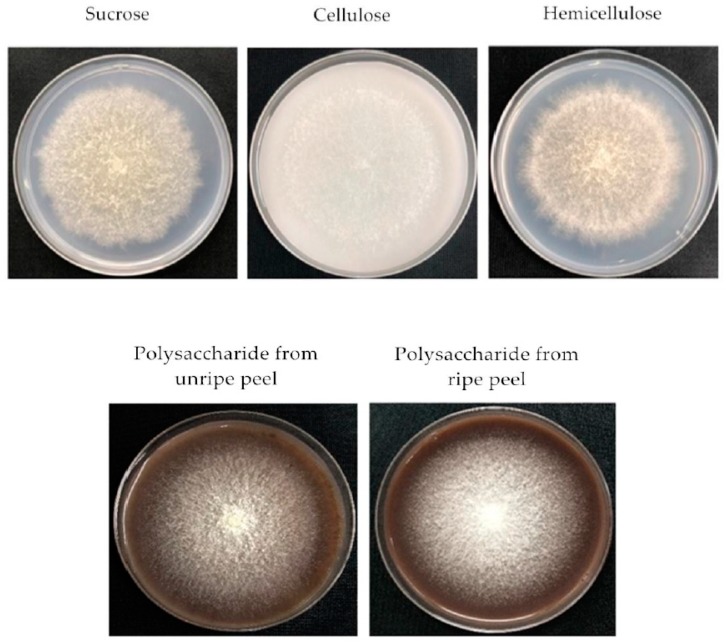
The morphology of *F. proliferatum* on the CB media with different carbon sources.

**Figure 4 toxins-11-00289-f004:**
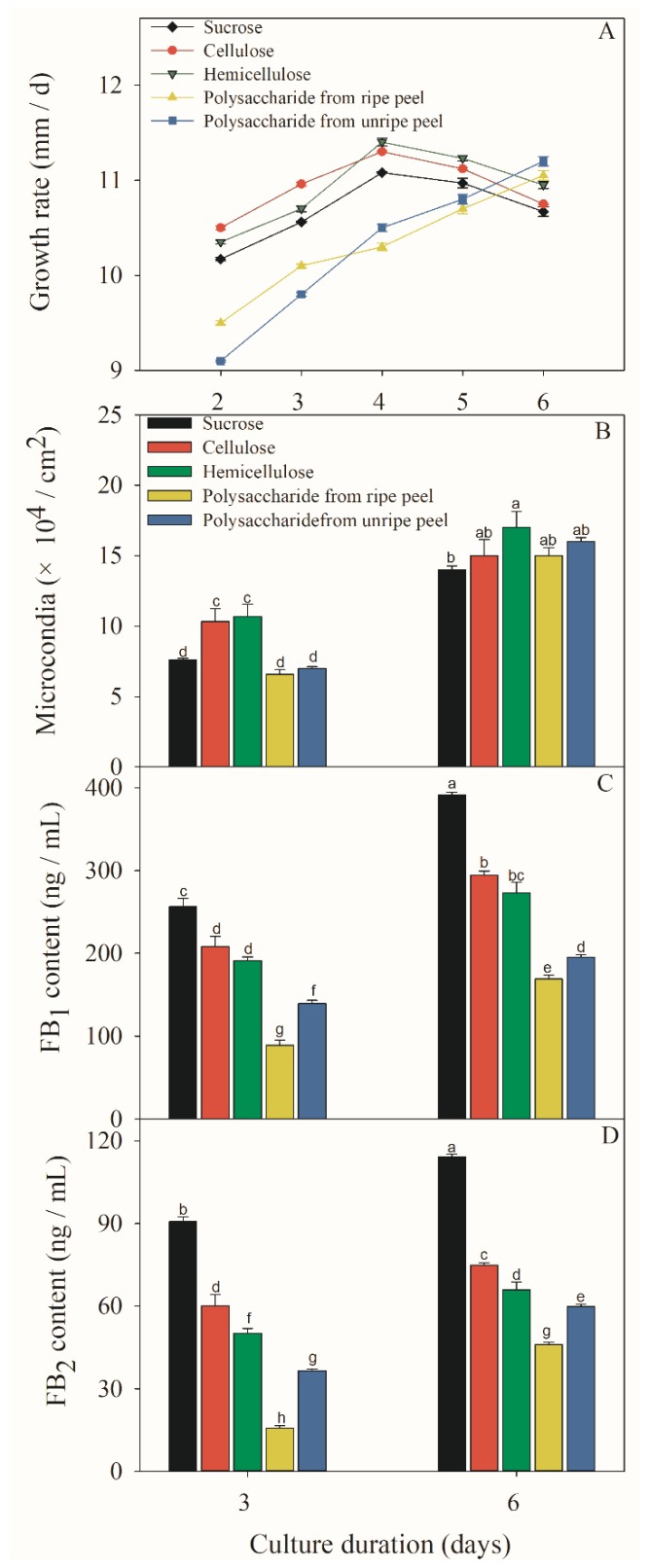
Effect of different sucrose conditions on fugal growth (**A**, **B**) and FB production (**C**, **D**) of *F. prolifeatum*. *F. proliferatum* were cultured in the culture media containing sucrose, cellulose, hemicellulose, and polysaccharide extracted from unripe or ripe banana peel for 6 days at 28 °C. The vertical bars indicate standard errors of three replicates. Different letters represent significant differences (*p* < 0.05).

**Figure 5 toxins-11-00289-f005:**
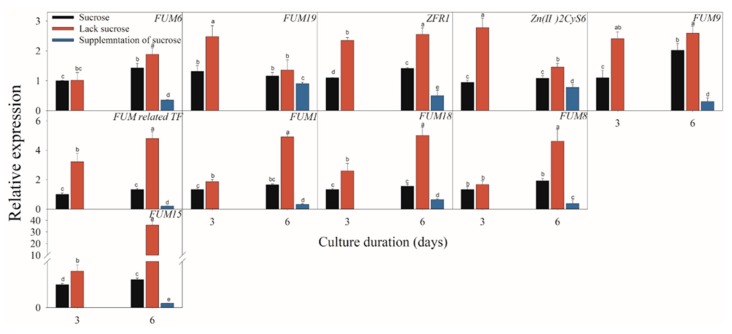
Effect of sucrose, lack sucrose and supplementation of sucrose on the expressions of the fumonisin-related genes of *F. proliferatum*. The detailed information of these genes is shown in Table 1. The data are presented as means of three independent replicates. The vertical bars indicate standard errors of three replicates. Different letters represent significant differences (*p* < 0.05).

**Figure 6 toxins-11-00289-f006:**
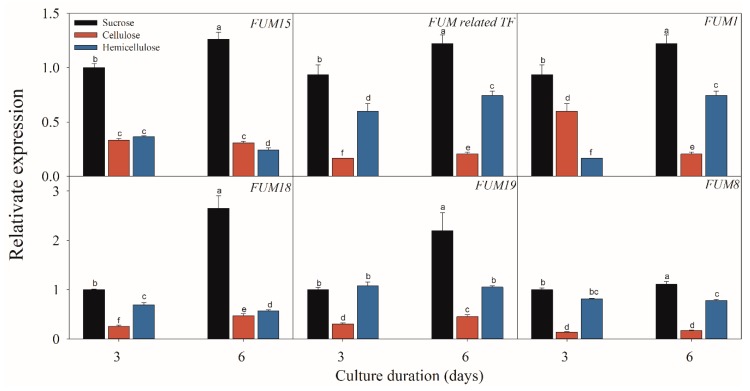
Effect of sucrose, cellulose and hemicellulose on the expressions of the fumonisin-related genes of *F. proliferatum*. The detailed information of these genes is shown in Table 1. The data are presented as means of three independent replicates. The vertical bars indicate standard errors of three replicates. Different letters represent significant differences (*p* < 0.05).

**Figure 7 toxins-11-00289-f007:**
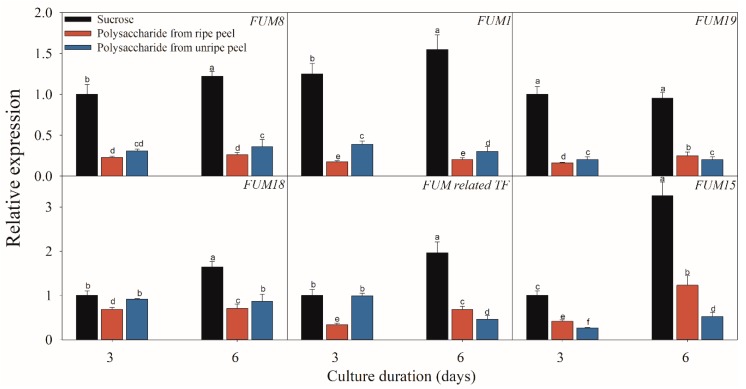
Effect of sucrose and polysaccharide extracted from unripe or ripe banana peel on the expressions of the fumonisin-related genes of *F. proliferatum*. The detailed information of these genes is shown in Table 1. The data are presented as means of three independent replicates. The vertical bars indicate standard errors of three replicates. Different letters represent significant differences (*p* < 0.05).

**Table 1 toxins-11-00289-t001:** Prime pairs used for RT-qPCR.

Gene	Description	Sequence of Primer (5’ to 3’)
*FUM1*	Polyketide synthase	For: ACTTTGCCATTTCCAACCGTAT
		Rev: GGGAGTTTTTCCATCCGAATTT
*FUM6*	Cytochrome P450 Monooxygenase	For: CGCTGGTACAGAAACGACGGCTAC
		Rev: TCGCGTAGGCACGCACTGAGATA
*FUM8*	Aminotransferase	For: ATTCCATGAGGAGGCAATGCAG
		Rev: GGTGCTATTCCTTCGAGGTCAC
*FUM9*	Dioxygenase	For: GAGCGTGGATGCTTGGCTGTTACT
		Rev: GGACTGGGAGCTTCTTTGCGGTATC
*FUM15*	Cytochrome P450 monooxygenase	For: CCATTCCACTCACGATGCGAGAAGC
		Rev: GCCAGGATTATTCTAGTGCCAGCAGGTA
*FUM18*	Longevity assurance factor	For: TGGTAGATGATGTGAGGAGCGACGA
		Rev: TCAAGTAGCCGTTGCCGTCATTCC
*FUM19*	ABC transporter	For: GGCTATGGATTCGGACGCTCTCAG
		Rev: ACCGTGCTGTGCTTGACCTAACATC
*FUM related TF*	Transcription factor	For: GCGGTGGAGGTGTCGGATTGAGTAA
		Rev: TGTCGGTGGAGGTAATGTAGTGGCTATTC
*ZFR1*	ZFR1 regulator of fumonisin biosynthesis	For: GCTCGTCTTCTCCTACATCGGCATCA
		Rev: CGGAATATGTGCGTTGTCAACAAGGTAGT
*FUM related Zn(II)2Cys6*	Fumonisin biosynthetic Related *Zn(II)2Cys6* protein	For: CAACTGCCAATAGCGAGGATGTGATGTC
		Rev: GACCTTCTCAACAATCCCGATTCCATTAC
*Histone H3*	*Histone H3*	For: ACTAAGCAGACCGCCCGCAGG
		Rev: GCGGGCGAGCTGGATGTCCTT

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
