# Peer review of "Effects of Different Carbon Sources on Fumonisin Production and FUM Gene Expression by Fusarium proliferatum"

_toxins, 2019, doi:10.3390/toxins11050289_

Round 1
Reviewer 1 Report
The overall motivation and context of this study must be stated more explicitly. The authors generally state that they would like to understand how differences in nutritional availability are associated with fungal growth and fumonisin production by F. proliferatum, which may have implications for how the fungus behaves to changing conditions during fruit ripening. Some major clarifications include:
-Why were banana peels and not other banana fruit tissues used in this study?
-Are the authors aiming to understand these phenomenon with respect the the banana-F. proliferatum pathosystem?
-If the authors are trying to understand more general phenomenon, why were other fruits not used and/or why not artificially mimic fruit ripening conditions?
The materials and methods should be described in greater detail. For example:
-How many biological (e.g. how many bananas from how many trees) and/or technical replicates were used?
-Briefly describe the fumonisin analysis and qPCR protocols from Li et al. (2017) and your previous publication. Not all readers or peer reviewers may have access to the literature sources cited in the materials and methods.
-Why was this gene set used for qPCR?
-Please describe your statistical methods in detail, including the full statistical models used, number of samples, and number of biological/technical replicates.
-Summaries of the statistical models should be included in the results section.
Author Response
The reviewer’s comments:
The overall motivation and context of this study must be stated more explicitly. The authors generally state that they would like to understand how differences in nutritional availability are associated with fungal growth and fumonisin production by F. proliferatum, which may have implications for how the fungus behaves to changing conditions during fruit ripening.
The author’s responses:
We thank the reviewer’s comments and suggestions for our manuscript. The manuscript has been revised ,as sguuested.
The reviewer’s comments:
Some major clarifications include:
Why were banana peels and not other banana fruit tissues used in this study?
The author’s responses:
We thank the reviewer’s careful review. During banana fruit–F. proliferatum interactions, the banana peel is the first natural infection place of F. proliferatum, and the F. proliferatum mainly grow on the banana peel after infection. Therefore, we used banana peel tissue in this study. Additionally, we also made corresponding revisions in the revised manuscript (Line 77-81).
The reviewer’s comments:
Are the authors aiming to understand these phenomenon with respect the banana - F. proliferatum pathosystem?
The author’s responses:
Thanks for reviewer’s comment. Banana, as one of the most economically important fruit crops worldwide, deteriorates easily due to rot development caused by postharvest pathogens, including F. proliferatum. Meanwhile, banana peel is rich in many sugar ingredients like cellulose, hemicellulose and pectin as the carbon sources when the fruit become edible and F. proliferatum might face these polysaccharides while infecting banana fruit. Hence, investigation of the effect of these polysaccharides on the fumonisin production in F. proliferatum might be favor to better understanding banana-F. proliferatum pathosystem. We also made corresponding revisions in the revised manuscript (Lines 73-74 and 77-81).
The reviewer’s comments:
If the authors are trying to understand more general phenomenon, why were other fruits not used and/or why not artificially mimic fruit ripening conditions?
The author’s responses:
Thanks for valuable suggestion. We agreed with the reviewer that using other fruits can be better understanding the general phenomenon. Unfortunately, as explained above, in this study, we mainly focused on banana fruit. In this study, we conducted the experiments in vitro scale to focus on the effect of different carbon sources on F. proliferatum growth and fumonisin biosynthesis. Under natural fruit ripening conditions, the environment factors were complicated and the effect does not depend on the sole effect from carbon sources. Therefore, we just used polysaccharide extracted from banana peel at different ripening stages (ripe and unripe) to mimic fruit ripening in vitro.
The reviewer’s comments:
The materials and methods should be described in greater detail. For example:
How many biological (e.g. how many bananas from how many trees) and/or technical replicates were used?
The author’s responses:
We thank the reviewer’s valuable suggestion. We have provided more information in the Materials and methods section in the revised manuscript (Lines 267, 261, 286 and 304).
The reviewer’s comments:
Briefly describe the fumonisin analysis and qPCR protocols from Li et al. (2017) and your previous publication. Not all readers or peer reviewers may have access to the literature sources cited in the materials and methods.
The author’s responses:
Thanks. We have provided more information on fumonisin analysis and qPCR protocols in the revised manuscript (Lines 278-278, 279-286 and 296-301).
The reviewer’s comments:
Why was this gene set used for qPCR?
The author’s responses:
Many thanks for the reviewer’s careful review and valuable suggestion. Considering that these genes were involved in fumonisin production, we found that the expression levels of these genes were regulated by different carbon sources (Li, et al., 2017, Carbon sources influence fumonisin production in Fusarium proliferatum, Proteomics). Hence, we also used these genes for qPCR in this study. In addition, we also made corresponding revisions in the revised manuscript (Lines 301-303).
The reviewer’s comments:
Please describe your statistical methods in detail, including the full statistical models used, number of samples, and number of biological/technical replicates.
The author’s responses:
Thank the reviewer’s valuable suggestion. As suggested, we have described the statistical methods in detail in the revised manuscript (Line 261, Line 286 and Line 304 and Line 308-310).
The reviewer’s comments:
Summaries of the statistical models should be included in the results section.
The author’s responses:
Thanks for the reviewer’s valuable suggestions. As suggested, we have made corresponding revisions in the revised manuscript (Lines 105, 107-108, 127, 128, 143-144, 151, 156 and 161).
Reviewer 2 Report
In the present study, the authors have checked the effect of carbon source in the growth of a common mycotoxigenic fungi, Fusarium proliferatum, in vitro and the fumonisin biosynthesis. Although the reported results are limited to the in vitro scale, it will be exciting to confirm these results on the plant which in this case is the banana fruit which can be a useful tool for biocontrol of some fungal disease in banana. The experimental design seems to be sound and the data analysis is convenient. The introduction is long and contains too many and sometimes inappropriate references. I would suggest some minor corrections. My suggestions are below.
Introduction
Ø Line 40, its fumonisins.. rewrite the sentence.
Ø Line 41, add reference.
Ø Line 46, I would move this sentence before 43. So mention the types then the IARC classification.
Ø Line 48, references 5 and 6 are not suitable to. Please replace them. I suggest the following Hove et al., 2016. Review on the natural co-occurrence of AFB1 and FB1 in maize and the combined toxicity of AFB1 and FB1.
Ø Line 50, replace reference number 7 with voss 2007, Fumonisins: toxicokinetics, mechanism of action and toxicity and Abdallah et al., 2015 Occurrence, Prevention and Limitation of Mycotoxins in Feeds.
Ø Line 53 too many references, replace by one review stating the infection of Fusarium proliferatum in these plants and crops.
Ø Line 56, the references 19 didn’t mention the presence of Fumonisins in Tomato ! double check all your references and source of your information.
Ø Line 68, reference number 30 is not appropriate. Its about A.s niger. I suggest to remove it.
Results and Discussion
Ø Line 90, replace the word “obtained” with “checked” or “measured”.
Ø For the figures, I would suggested to make them bigger and in color for a better visibility to the readers.
Author Response
The reviewer’s comments:
In the present study, the authors have checked the effect of carbon source in the growth of a common mycotoxigenic fungi, Fusarium proliferatum, in vitro and the fumonisin biosynthesis. Although the reported results are limited to the in vitro scale, it will be exciting to confirm these results on the plant which in this case is the banana fruit which can be a useful tool for biocontrol of some fungal disease in banana. The experimental design seems to be sound and the data analysis is convenient. The introduction is long and contains too many and sometimes inappropriate references. I would suggest some minor corrections. My suggestions are below.
The author’s responses:
We thank very much the reviewer for the positive evaluation of our work. These suggestions have been used to improve the revised manuscript.
Introduction
The reviewer’s comments:
Line 40, its fumonisins. rewrite the sentence.
The author’s responses:
Thanks. We have changed fumonisin into fumonisins, and then rewrite the sentence in the revised manuscript (Line 40).
The reviewer’s comments:
Line 41, add reference.
The author’s responses:
We thank the reviewer’s careful review. We have added references into the revised manuscript (Lines 41 and 322-323).
The reviewer’s comments:
Line 46, I would move this sentence before 43. So mention the types then the IARC classification.
The author’s responses:
We thank the reviewer’s suggestion. We have made corresponding revision in the revised manuscript (Lines 42-43).
The reviewer’s comments:
Line 48, references 5 and 6 are not suitable to. Please replace them. I suggest the following Hove et al., 2016. Review on the natural co-occurrence of AFB1 and FB1 in maize and the combined toxicity of AFB1 and FB1.
The author’s responses:
We thank the reviewer’s valuable suggestion. We have made corresponding revisions in the revised manuscript, as suggested (Lines 49 and 332-334).
The reviewer’s comments:
Line 50, replace reference number 7 with voss 2007, Fumonisins: toxicokinetics, mechanism of action and toxicity and Abdallah et al., 2015 Occurrence, Prevention and Limitation of Mycotoxins in Feeds.
The author’s responses:
Many thanks for reviewer’s suggestion and we have made corresponding revisions as suggested in the revised manuscript (Lines 51, 335-338 and 339-340).
The reviewer’s comments:
Line 53 too many references, replace by one review stating the infection of Fusarium proliferatum in these plants and crops.
The author’s responses:
Thanks for the valuable suggestion. We have replaced these references by one review (Summerell et al., Biogeography and phylogeography of Fusarium: a review. Fungal Divers, 2010, 44, 3-13, doi:10.1007/s13225-010-0060-2). We have also made corresponding revisions in the revised manuscript (Lines 54 and 343-344).
The reviewer’s comments:
Line 56, the references 19 didn’t mention the presence of Fumonisins in Tomato! double check all your references and source of your information.
The author’s responses:
Many thanks for the reviewer’s careful review. We have corrected them in the revised manuscript (Lines 56 and 357-358). We have checked all the references and made corrections in the revised manuscript.
The reviewer’s comments:
Line 68, reference number 30 is not appropriate. Its about A.s niger. I suggest to remove it.
The author’s responses:
Many thanks for the reviewer’s careful review. We have removed it in the revised manuscript (Line 68).
Results and Discussion
The reviewer’s comments:
Line 90, replace the word “obtained” with “checked” or “measured”.
The author’s responses:
Thanks for the valuable suggestion. We have replaced the word “obtained” with “checked” in the revised manuscript (Line 98).
The reviewer’s comments:
For the figures, I would suggested to make them bigger and in color for a better visibility to the readers.
The author’s responses:
Many thanks for reviewer’s suggestion. We have made corresponding revisions in the revised manuscript (Lines 113, 132, 147, 152 and 157).